# Preventing Postoperative Catheter-Related Bladder Discomfort (CRBD) with Bladder Irrigation Using 0.05% Lidocaine Saline Solution: Monitoring with Analgesia Nociception Index (ANI) after Transurethral Surgery

**DOI:** 10.3390/medicina60091405

**Published:** 2024-08-27

**Authors:** Chia-Heng Lin, I-Cheng Lu, Tz-Ping Gau, Kuang-I Cheng, Hsin-Ling Chen, Ping-Yang Hu

**Affiliations:** 1Department of Anesthesiology, Kaohsiung Medical University Hospital, Kaohsiung 807377, Taiwanu9401066@gap.kmu.edu.tw (T.-P.G.);; 2School of Medicine, College of Medicine, Kaohsiung Medical University, Kaohsiung 807378, Taiwan

**Keywords:** catheter-related bladder discomfort (CRBD), bladder irrigation, lidocaine, analgesia nociception index (ANI), transurethral surgery

## Abstract

(1) *Background and Objectives*: Catheter-related bladder discomfort (CRBD), a common and distressing consequence of indwelling urinary catheters, can significantly impact postoperative recovery. This study aimed to determine the effectiveness of bladder irrigation with a 0.05% lidocaine normal saline solution for the prevention of CRBD following transurethral surgery. (2) *Materials and Methods*: In this randomized, double-blind, placebo-controlled trial, patients were assigned to either a control group receiving normal saline or a treatment group receiving 0.05% lidocaine (2% lidocaine 25 mL in 1000 mL saline) for bladder irrigation. Both groups were administered fentanyl (1 μg/kg) for analgesia at the end of the procedure. The primary endpoint was the assessment of the incidence and severity of CRBD upon awakening within the first 6 h postoperatively, using a four-grade scale based on the patients’ reports of discomfort. (3) *Results*: Out of 79 patients completing the study, the incidence of moderate to severe CRBD was significantly lower in the lidocaine group (5.1%, 2/39) compared to the control group (25%, 10/40) at 10 min after waking from anesthesia (*p* = 0.014). Furthermore, the lidocaine group experienced significantly less CRBD at 1 and 2 h postoperative (2.6% and 0%, respectively) compared to the control group (20% and 10%, respectively) (*p* = 0.015, *p* = 0.043), with no significant differences at 6 h (*p* = 0.317). (4) *Conclusions*: The results suggest that bladder irrigation with 0.05% lidocaine reduces the occurrence of moderate to severe CRBD by nearly 80% in the initial 2 h postoperative period after transurethral surgery.

## 1. Introduction

Catheter-related bladder discomfort (CRBD) is a postoperative complication in patients undergoing transurethral surgeries, manifesting as an urgent sensation of needing to void the bladder or discomfort in the suprapubic area due to the presence of a urinary catheter [1,2]. The pathophysiology of CRBD is primarily attributed to bladder irrigation, which might irritate the bladder mucosa and activate the muscarinic receptors. The insertion of a urinary catheter can irritate the bladder’s mucosal lining, leading to discomfort and an urge to urinate. No matter whether the irritation comes from a urinary catheter or irrigated normal saline, either can trigger the bladder’s sensory pathways, contributing to the sensation of discomfort and urgency. The bladder’s detrusor muscle, which contracts to expel urine, is regulated by muscarinic acetylcholine receptors. Among these, the M2 and M3 subtypes are particularly important in mediating bladder contractions. The M3 receptors are directly responsible for detrusor muscle contraction, leading to the bladder emptying, while the M2 receptors, although less abundant in the human bladder, can also contribute to detrusor contractions and have been implicated in the pathophysiology of various bladder disorders, including CRBD [3,4]. Involuntary bladder contractions induced by the activation of these receptors, especially in the context of bladder irritation and the presence of a catheter, can lead to symptoms associated with CRBD, including discomfort, urgency, and the sensation of needing to urinate.

The management of CRBD poses a significant challenge in postoperative care, with current strategies ranging from a reduction in the balloon volume of the indwelling urinary catheter to topical or systemic medications [5,6,7]. Various pharmacological agents including anticholinergics, analgesics (e.g., tramadol, paracetamol), antiepileptics (e.g., gabapentin, pregabalin) and anesthetics (e.g., ketamine, dexmedetomidine) have been explored for the prevention and treatment of CRBD [6,8,9], and despite the adequate efficacy of systemic medications, these drugs often present side effects such as facial flushing, dry mouth, blurred vision, and sedation.

This has led to the exploration of novel methods for CRBD management, including the use of local anesthetics such as lidocaine via either intravenous or bladder irrigation [10,11]. Lidocaine, by blocking the sodium channels in the bladder mucosa, reduces the afferent signaling that contributes to the sensation of discomfort. This mechanism of action provides a direct, targeted approach to alleviating CRBD, offering potential advantages in terms of its efficacy and side effect profile compared to systemic treatments.

Nociception, the physiological neural processing of painful stimuli, is closely linked to the autonomic nervous system. Painful stimuli lead to an increase in sympathetic activity and a decrease in parasympathetic tone. By analyzing the changes in heart rate variability, which reflect the balance between sympathetic and parasympathetic activity, the Analgesia Nociception Index (ANI) aims to provide an objective measure of nociception. Recent advancements have introduced the Analgesia Nociception Index (ANI) as a non-invasive tool to monitor the balance between nociception and anti-nociception, offering a real-time assessment of a patient’s analgesic status [12,13].

This study aimed to explore the efficacy of a 0.05% lidocaine saline solution for bladder irrigation to reduce the incidence and severity of CRBD, with the ANI serving as a monitoring tool to optimize analgesic management. We anticipate that this study could provide not only enhanced patient comfort but also align well with the principles of personalized medicine by tailoring interventions to individual patients’ needs.

## 2. Materials and Methods

This randomized, double-blind, placebo-controlled study was approved by the Institutional Review Board of Kaohsiung Medical University Hospital (KMUHIRB-F(I)-20190065), and the trial was registered at ClinicalTrials.gov, with the clinical trial number stated here (identifier: NCT04133571). Informed consent was obtained for each patient at the preoperative counseling clinic. We conducted this trial for patients undergoing elective transurethral surgery under general anesthesia between May 2019 and December 2022. Exclusion criteria in this study were being aged ≤ 20 and ≥80 years, or having severe cardiovascular, pulmonary, hepatic, or renal disease.

### 2.1. Anesthesia Protocol

All patients were fasted for at least 8 h before general anesthesia, and no medications were administered. Each patient received standard monitoring, including electrocardiography (lead II), noninvasive BP testing, pulse oximetry, and end-tidal carbon dioxide (EtCO2) measurement. Patients were preoxygenated with 6 L/min 100% oxygen via a face mask to achieve peripheral oxygen saturation of near 100% before induction. Anesthesia was induced with 0.5 μg/kg fentanyl, 3 mg/kg thiamylal, and 1 mg/kg propofol. A #3 or #4 laryngeal mask airway was placed on the female and male patients, respectively, while anesthesia was maintained with sevoflurane 1~1.3 MAC with an oxygen flow of 0.3 L/min and mixed air of 0.7 L/min. Patients were allocated into two groups according to their bladder irrigation solution after transurethral surgery; patients in the control group received normal saline for bladder irrigation for 30 min in the PACU, while patients in the lidocaine group received a 0.05% lidocaine normal saline solution. Both groups were given 1 mcg/kg fentanyl for analgesics at the end of surgery. All patients were sent to the postoperative care unit (PACU) for further care.

### 2.2. Outcome Measurements

The primary outcome of the study was the incidence and severity of CRBD after transurethral surgery in the PACU. The bladder discomfort score was assessed when the patient awakened in the PACU and again at 10 min, 20 min, 40 min, 1 h, 2 h, and 6 h postoperatively in the urology ward by a registered nurse who was blinded to the patient group. The bladder discomfort was graded on a CRBD scale divided into 4 grades: Grade 0, none: patients did not complain of any CRBD even on asking; Grade 1, mild: reported by the patient only on asking; Grade 2, moderate: reported by the patient on their own (the patient subjectively reported this, but it was not accompanied by any behavioral responses); Grade 3, severe: reported by the patient on their own along with behavioral responses (behavioral responses observed were flailing limbs, a strong vocal response, and attempts to pull out the catheter). Patients who exhibited high-grade CRBD, identified as Grade 2 or 3, were administered intravenous 0.25 mg/kg meperidine as a rescue therapy. In cases where patients presented with low-grade CRBD, identified as Grade 0 or 1, we opted for observation without intervention.

Secondary outcomes included the consciousness level, ANI score, and numbers needing rescue. The consciousness level was assessed by a sedation scale as per the following: Grade 0: fully awake; Grade 1: drowsy, but easily woken from reaction to voice; Grade 2: drowsy, hardly able to wake, only responding to pain; Grade 3: unconsciousness. The ideal ANI range was 50–70, where an ANI value below 50 was considered inadequate analgesia. We also recorded comprehensive safety assessments including postoperative nausea and vomiting (PONV), dizziness, allergic reactions, and any systemic toxicity of lidocaine (tongue tingling, metallic taste, or seizure)**.**

### 2.3. Statistical Analysis

The sample size was determined based on an anticipated incidence of 47.5% in the primary outcome, which corresponds to moderate and severe catheter-related bladder discomfort (CRBD) [14]. The estimated decrease in CRBD was 30%. A statistical power of 80% and a significance level (alpha) of 0.05 were used. A two-tailed test was chosen based on the bi-directional nature of our hypothesis. Based on these assumptions, the required sample size was calculated to be 37 participants per group, accounting for a potential dropout rate of 5%, leading to a final sample size of 40.

Demographic data were collected and are presented as mean and standard deviation (SD); continuous data were presented as mean ± standard deviation (SD) values, and categorical data were compared using the Chi-squared test or Fisher’s exact test, as appropriate, with the results expressed as numbers (%). All statistical analyses were conducted using the package SPSS 18.0. *p* < 0.05 was considered significant.

## 3. Results

A total of 80 patients undergoing elective transurethral surgery under general anesthesia during the research period were invited to participate in the study, with one case in the lidocaine group with discontinued intervention being excluded. The remaining 79 cases (control group N = 40, lidocaine group N = 39) were completely analyzed (Figure 1). The demographic and patient characteristics are summarized in Table 1. There was no significant difference between the groups.

The primary outcome is presented in Table 2, showing that the patients in the lidocaine group experienced significantly fewer instances of high-grade CRBD compared to those in the control group at 10, 60 and 120 min after awakening (all *p* < 0.05).

The consciousness level and ANI scores within an hour in the PACU are presented in Table 3 and Table 4, respectively. Postoperative drowsiness did not reveal a difference between the groups (*p* > 0.05). As recovery progressed, the percentage of fully awake (Grade 1) patients increased in both groups, with the majority of patients being fully awake after 60 min in the PACU (82.5% in the control group and 76.9% in the lidocaine group). None of the patients in either group experienced a coma (Grade 4) after 10 min in the PACU.

The ANI scores for both groups at each time interval are relatively close, without a significant difference. At 20 min, the lidocaine group showed a higher average score (67.10) compared to the control group (62.48), but the difference was not statistically significant (*p* = 0.232). Table 4 provides a comparison of the ANI scores between groups, indicating that lidocaine administration did not significantly impact the ANI scores at various time intervals post-operation.

Table 5 presents a significant reduction in rescue cases in the lidocaine group (1 of 39) compared to the control group (7 of 40) (*p* = 0.028). There were no significant differences between the groups in terms of operation time (*p* = 0.252) or specimen weight (*p* = 0.803). The incidence of PONV was 2 out of 39 in the lidocaine group and 1 out of 40 in the control group (*p* = 0.982). Dizziness was reported in 2 out of 39 in the lidocaine group and 0 out of 40 in the control group (*p* = 0.463). No patients in either group exhibited any allergic reactions or symptoms of lidocaine toxicity.

## 4. Discussion

The research provided postoperative low-concentration lidocaine bladder irrigation to alleviate bladder discomfort for patients undergoing transurethral surgery, and the main results showed that the incidence of high-grade (moderate to severe) CRBD was significantly lower in the lidocaine group (5.1%, 2/39) compared to the control group (25%, 10/40) at 10 min after awakening from anesthesia, with its effect lasting for 120 min. Additionally, fewer cases in the lidocaine group (2.5%, 1/39) required rescue meperidine compared to the control group (17.5%, 7/40). To our knowledge, only a limited number of randomized controlled trials (RCTs) have assessed the effect of lidocaine irrigation on CRBD [15].

The significant reduction in high-grade CRBD observed in patients with 0.05% lidocaine (2% lidocaine 25 mL in 1000 mL saline) highlights the efficacy of local anesthetic bladder irrigation as a targeted approach for postoperative discomfort management in patients undergoing transurethral surgery. Our results delineated the high-grade CRBD from 25% in the control group to 5.1% in the lidocaine group at the time interval of awakening at 10 min. This indicated a nearly 80% decrease in high-grade CRBD using lidocaine instead of normal saline irrigation postoperatively.

Lidocaine application through a Foley catheter in urological procedures has been shown to enhance patient comfort and reduce the need for additional analgesics. Singh et al. reported a 52% reduction in high-grade (II-III) CRBD incidence by transurethral irrigation with 0.01% lidocaine (2% lidocaine 5 mL in 1000 mL saline) for 30 min before the end of transurethral resection of bladder tumors (TURBT), with the analgesic effect of 0.01% lidocaine irrigation persisting within 2 h [11]. However, with 0.05% lidocaine irrigation in our study, adequate analgesia lasting for 120 min postsurgically was obtained. The higher concentration of lidocaine (0.05%) irrigation might be a better regimen with a higher reduction in CRBD incidence.

As a local anesthetic, lidocaine numbs the bladder mucosa by maintaining the sodium channels in inactivated states on the cell membranes, thereby inhibiting the conduction of nerve impulses that signal pain from the bladder to the brain [16,17]. This mechanism of lidocaine provides a direct and targeted method to alleviate bladder discomfort associated with the indwelling urinary catheter. The route of administered local anesthetic is particularly beneficial when continuous bladder irrigation is necessary after transurethral surgery settings such as transurethral resection of bladder tumors (TURBT), transurethral resection of the prostate (TURP), or ureterorenoscopic lithotripsy (URSL).

Under safe margin, duration and concentration are the key factors in choosing local anesthetics for bladder irrigation. According to previous reports, CRBD is exacerbated strongly in postoperative patients within a one-hour period and persisting for several hours; however, while examining the grading of *yes or no* items at 6 h post-operation, the majority of patients’ CRBD symptoms subsided due to the use of either conventional medications or experimental medicines [10]. Though local anesthetics with longer durations, such as levobupivacaine or ropivacaine, might provide longer attenuation to CRBD symptoms [18,19], we selected the widely used lidocaine with a short duration in this study. With respect to the CRBD time course, the lidocaine group experienced significantly fewer CRBD symptoms at 1 and 2 h postoperatively (2.6% and 0%, respectively) compared to the control group (20% and 10%, respectively) (*p* = 0.015, *p* = 0.043). High-grade CRBD reached 7.5% without intervention and 2.5% with irrigation at 6 h postoperatively (*p* = 0.317).

Hence, lidocaine with a relatively short duration for irrigation might be adequate. Pehlivanoğlu et al. reported pain score by visual analog scale (VAS) after cystoscopies under local anesthesia by bladder infusion with levobupivacaine solution diluted to 30 mL, where mean VAS score was 2.2 and 1.45 for 0.5% levobupivacaine at 4 and 8 mL, respectively (*p* < 0.01) [20], suggesting that higher concentrations of local anesthetics might be necessary for bladder infusion to relieve CRBD. Nevertheless, the minimal effective concentration of local anesthetic irrigation remains to be determined and deserves future investigation.

The absence of significant differences in consciousness levels and ANI scores between the two groups suggested the localized action of lidocaine without systemic effects on sedation or overall pain perception. The ANI measures the balance between parasympathetic and sympathetic nervous system activity, reflecting the pain response and the effectiveness of analgesia. By monitoring heart rate variability (HRV) as a proxy, ANI provides a non-invasive, real-time assessment of the analgesia/nociception balance during anesthesia, aiming to optimize pain management by indicating when additional analgesia might be needed to counteract nociceptive pain [21]. The ANI is specifically designed to correlate with nociceptive pain, which arises from tissue damage triggering nociceptors rather than transmitting signals to the central nervous system, while visceral pain originates from the internal organs i.e., bladder discomfort. This might explain the absence of an ANI score correlation in this study. The rationale for intraoperative and postoperative ANI monitoring includes the following points: First, ANI provides a non-invasive, real-time assessment of the analgesia/nociception balance during anesthesia, offering valuable insights even if it does not directly correlate with visceral pain. Second, there are limited alternative options for objectively monitoring nociception in anesthetized patients, and ANI is one of the more promising tools available. Third, ANI may still indirectly reflect changes in nociception related to bladder discomfort, even if it is not a direct measure of visceral pain. Despite the apparent discrepancy between ANI findings and CRBD, the use of ANI suggested that patients in both groups were under comparable nociceptive pain control from anesthesia induction to recovery in the PACU.

## 5. Conclusions

Lidocaine bladder irrigation effectively reduced the occurrence of moderate to severe CRBD in the initial 2 h postoperative period after transurethral surgery. It is a practical method with minimal side effects when compared to systemic medications such as anticholinergics or opioids for CRBD in the early postoperative hours.

## Figures and Tables

**Figure 1 medicina-60-01405-f001:**
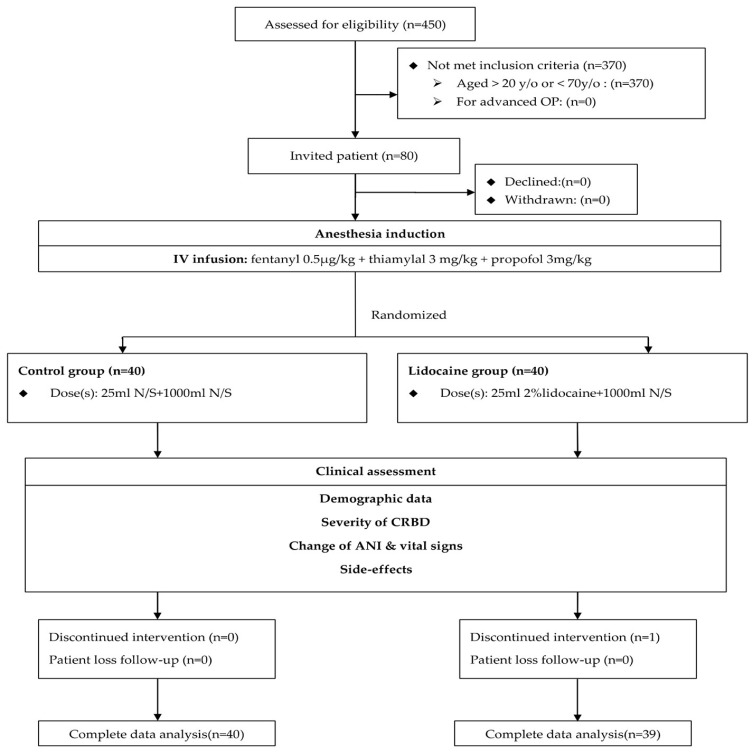
Patient flow chart. Control and lidocaine groups represented bladder irrigation with saline and lidocaine, respectively. CRBD = catheter-related bladder discomfort; ANI = Analgesia Nociception Index.

**Table 1 medicina-60-01405-t001:** Demographics and characteristics data.

Variables	Control (n = 40)	Lidocaine (n = 39)	*p*-Value
Age (y/o)	63.5 ± 6.9	63.9 ± 6.6	0.820
Height (cm)	166.9 ± 5.7	167.4 ± 5.5	0.672
Weight (kg)	70.9 ± 12.0	69.4 ± 8.2	0.523
BMI (kg/m^2^)	25.5 ± 3.7	24.8 ± 2.9	0.338
Diabetes	7 (17.5%)	9 (23.1%)	0.537
Hypertension	9 (22.5%)	17 (43.6%)	0.046
Asthma	0 (0%)	1 (2.6%)	0.308
CVA	1 (2.5%)	1 (2.6%)	0.986
ASA (II/III)	31/9	27/12	0.406

BMI = body mass index; CVA = cerebral vascular accident; ASA = American Society of Anesthesiologists physical status classification.

**Table 2 medicina-60-01405-t002:** Postoperative catheter-related bladder discomfort (CRBD) in transurethral surgery patients.

Variables	Control Group (n = 40)	Lidocaine Group (n = 39)	*p*-Value
Awake			
Low	35 (87.5%)	38 (97.4%)	0.096
High	5 (12.5%)	1 (2.6%)	
Awake 10 min			
Low	30 (75.0%)	37 (94.9%)	0.014
High	10 (25.0%)	2 (5.1%)	
Awake 60 min			
Low	32 (80.0%)	38 (97.4%)	0.015
High	8 (20.0%)	1 (2.6%)	
Awake 120 min			
Low	36 (90.0%)	39 (100%)	0.043
High	4 (10.0%)	0 (0%)	
Awake 360 min			
Low	37 (92.5%)	38 (97.4%)	0.317
High	3 (7.5%)	1 (2.6%)	

Low = CRBD grade 0 or 1; High = CRBD grade 2 or 3.

**Table 3 medicina-60-01405-t003:** Postoperative consciousness level in transurethral surgery patients.

Variables	Control Group (n = 40)	Lidocaine Group (n = 39)	*p*-Value
**PACU Arrival**			
Grade 1	17 (42.5%)	11 (28.2%)	0.339
Grade 2	22 (55.0%)	24 (61.5%)	
Grade 3	1 (2.5%)	3 (7.7%)	
Grade 4	0 (0.0%)	1 (2.6%)	
**PACU 10 min**			
Grade 1	27 (67.5%)	19 (48.7%)	0.233
Grade 2	12 (30.0%)	18 (46.2%)	
Grade 3	1 (2.5%)	2 (5.1%)	
Grade 4	0 (0.0%)	0 (0.0%)	
**PACU 20 min**			
Grade 1	29 (72.5%)	25 (64.1%)	0.714
Grade 2	10 (25.0%)	13 (33.3%)	
Grade 3	1 (2.5%)	1 (2.6%)	
Grade 4	0 (0.0%)	0 (0.0%)	
**PACU 40 min**			
Grade 1	29 (72.5%)	27 (69.2%)	0.749
Grade 2	11 (27.5%)	12 (30.8%)	
Grade 3	0 (0.0%)	0 (0.0%)	
Grade 4	0 (0.0%)	0 (0.0%)	
**PACU 60 min**			
Grade 1	33 (82.5%)	30 (76.9%)	0.473
Grade 2	7 (17.5%)	9 (23.1%)	
Grade 3	0 (0.0%)	0 (0.0%)	
Grade 4	0 (0.0%)	0 (0.0%)	

PACU = postoperative care unit. Grade 1: fully awake; Grade 2: sleepy, easy to wake up; Grade 3: sleepy and difficult to wake up with pain; Grade 4: coma.

**Table 4 medicina-60-01405-t004:** Postoperative analgesia nociception index (ANI) level in transurethral surgery patients.

Variables	Control Group (n = 40)	Lidocaine Group (n = 39)	*p*-Value
PACU	65.31 ± 11.95	64.87 ± 15.30	0.889
PACU 10 min	65.05 ± 16.03	64.05 ± 16.99	0.789
PACU 20 min	62.48 ± 16.28	67.10 ± 17.80	0.232
PACU 40 min	64.15 ± 17.52	65.28 ± 16.43	0.770
PACU 60 min	64.05 ± 17.61	65.53 ± 17.65	0.715

PACU = postoperative care unit.

**Table 5 medicina-60-01405-t005:** Rescue cases and operation profile.

Variables	Control Group (n = 40)	Lidocaine Group (n = 39)	*p*-Value
Rescue No	33 (82.5%)	38 (97.4%)	0.028
Yes	7 (17.5%)	1 (2.6%)	
Operation time	75.35 ± 55.03	88.08 ± 41.79	0.252
Specimen weight	20.93 ± 17.26	19.16 ± 17.99	0.803

Rescue was performed by intravenous meperidine 0.25 mg/kg in the PACU.

## Data Availability

Available data are contained in tables in in the manuscript.

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
