# Peer review of "Preventing Postoperative Catheter-Related Bladder Discomfort (CRBD) with Bladder Irrigation Using 0.05% Lidocaine Saline Solution: Monitoring with Analgesia Nociception Index (ANI) after Transurethral Surgery"

_medicina, 2024, doi:10.3390/medicina60091405_

Round 1
Reviewer 1 Report
Comments and Suggestions for Authors
This randomized, placebo-controlled study evaluates the effectiveness of 0.05% lidocaine saline bladder irrigation in reducing CRBD in patients post-transurethral surgery. Results show a significant reduction in moderate to severe CRBD in the lidocaine group compared to the control group. Methods are quite robust, with ANI used to monitor analgesia. While results are sound and discussion balanced, 1) the study would benefit from extended follow-up for long-term efficacy and detailed methodology on ANI utilization. 2) this is an established practice in numerous European institutions
Author Response
Comment-1:
This randomized, placebo-controlled study evaluates the effectiveness of 0.05% lidocaine saline bladder irrigation in reducing CRBD in patients post-transurethral surgery. Results show a significant reduction in moderate to severe CRBD in the lidocaine group compared to the control group. Methods are quite robust, with ANI used to monitor analgesia. While results are sound and discussion balanced, 1) the study would benefit from extended follow-up for long-term efficacy and detailed methodology on ANI utilization. 2) this is an established practice in numerous European institutions
Response:
Thank you so much for pointing out this important issue.
1-1. Extended Follow-Up for Long-Term Efficacy:
We appreciate the suggestion to include extended follow-up for long-term efficacy. We acknowledge that long-term data could provide valuable insights into the sustained effects of lidocaine bladder irrigation on CRBD. We shall arrange future studies to incorporate follow-up periods.
1-2. Detailed Methodology on ANI Utilization:
ANI monitor was based on relationship between nociception and autonomic nervous system. We have added the rational of methodology on ANI use in the introduction section.
Manuscript Changes:
Nociception, the physiological neural processing of painful stimuli, is closely linked to the autonomic nervous system. Painful stimuli lead to an increase in sympathetic activity and a decrease in parasympathetic tone. By analyzing changes in heart rate variability, which reflects the balance between sympathetic and parasympathetic activity, ANI aims to provide an objective measure of nociception.
- Established Practice in European Institutions:
We also appreciate the use of lidocaine bladder irrigation in numerous European institutions. Our results may provide clinical evidence to support this treatment. This contextual information reinforces the relevance and applicability of our findings within a global framework.

Reviewer 2 Report
Comments and Suggestions for Authors
The manuscript offers valuable insights into the management of CRBD, but there are several key areas needing improvement. The sample size determination and the rationale for using ANI require clearer explanation, while visual data presentation could significantly improve the interpretation of results. Additionally, consistency in terminology and formatting, as well as a detailed summary of safety assessments, will strengthen the manuscript's impact and readability.
Major comments:
1. I find the sample size determination in Section 2.3 Statistical Analysis difficult to follow. Can you please rewrite the paragraph for better clarity? For example, the following standard text would be a good starting point: “The sample size was determined based on an expected difference of [insert expected difference] in [primary outcome] with an estimated standard deviation of [insert standard deviation]. A statistical power of [insert power, e.g., 80%] and a significance level (alpha) of [insert significance level, e.g., 0.05] were used. A [one-/two-]tailed test was chosen based on [reason for choice, e.g., the directionality of the hypothesis]. Based on these assumptions, the required sample size was calculated to be [insert number] participants per group, accounting for a potential dropout rate of [insert dropout rate], leading to a final sample size of [insert final sample size].” For further guidance, you might look at Eng J. Sample size estimation: how many individuals should be studied? Radiology. 2003 May;227(2):309-13 for more details.
2. Table 2 is not very intuitive for interpreting the study results. While it's acceptable to note in the text that there is a significant difference between low- and high-grade CRBD in the two groups, for a primary endpoint, I would expect to see the scores for each grade (Grade 0, 1, 2, and 3) at different time points displayed in one or multiple graphs. This visual presentation would be more effective than a table.
3. The authors explain that “The analgesia nociception index (ANI) is specifically designed to correlate with nociceptive pain” and then clarify that bladder discomfort is classified as “visceral pain” rather than nociceptive pain. This, they suggest, might explain the lack of ANI score correlation in their study. To me, this indicates that ANI is not an appropriate tool for monitoring nociception in cases of bladder discomfort. It is then unclear why ANI was chosen for this study. Could the authors please elaborate on the rationale behind using ANI despite this apparent discrepancy in their Discussion section?
4. I suggest that the authors provide a summary of the safety assessments conducted in their study and discuss their findings in detail. Specifically, did they monitor for symptoms such as nausea, vomiting, dizziness, tongue tingling, metallic taste, allergic reactions, and so on? If these assessments were performed, what were the incidence rates in the two treatment groups, and were there significant differences? This information is essential to substantiate the authors' conclusion that “Lidocaine bladder irrigation [...] is a practical method with minimal side effects.”
Minor comments:
1. Please include additional information on the primary endpoint, catheter-related bladder discomfort (CRBD), in the Abstract. For example: “The primary endpoint was the assessment of the incidence and severity of CRBD upon awakening within the first 6 hours postoperatively, using a four-grade scale based on patients' reports of discomfort.”
2. In the Introduction, please correct the following sentence: “The pathophysiology of CRBD is indeed primarily attributed to the bladder irrigation might irritate the bladder mucosa and activate the muscarinic receptors.” I believe it should read: “The pathophysiology of CRBD is primarily attributed to bladder irrigation, which might irritate the bladder mucosa and activate the muscarinic receptors.”
3. In the background subsection of the Abstract, the drug name "lidocaine" is written with a capital letter as “0.05% Lidocaine”. Please correct it to “0.05% lidocaine”.
4. In Section 2.2 Outcome Measurements, "catheter-related bladder discomfort" is written out, even though the abbreviation "CRBD" has been previously used. The same applies to “analgesia nociception index (ANI)” at different places throughout the manuscript.
5. In the Discussion section, it is mentioned that “To our knowledge, only limited RCTs assessed the effect of lidocaine irrigation on CRBD [15].” Do the authors mean “[…] only a limited number of RCTs”? If not, please clarify why the referenced RCT (number 15 in the reference list) is of limited quality.
6. The citations for the works of Kim et al. and Agarwal et al. are incorrectly formatted in the Discussion section: “(citing reference attached Doo-Hwan Kim, 2020; A. Agarwal, 2008;)”. Please correct this.
7. There are inconsistencies in the use of spacing before units. For example, incorrect forms such as “20mins”, “25ml”, and “1μg/kg” are used, while correct spacing is seen in “80 years”, “6 h”, and “0.5 mcg/kg”. Please ensure a space is inserted before the unit throughout the manuscript. Additionally, please use either “μg” or “mcg” consistently.
Comments on the Quality of English LanguageThere are inconsistencies in terminology and formatting that need to be addressed. Additionally, the authors sometimes use articles like 'an' and 'a' incorrectly, either including them unnecessarily or omitting them where they are required. It is recommended that a skilled English writer proofread the manuscript for these errors.
Author Response
Major comments:
- I find the sample size determination in Section 2.3 Statistical Analysis difficult to follow. Can you please rewrite the paragraph for better clarity? For example, the following standard text would be a good starting point: “The sample size was determined based on an expected difference of [insert expected difference] in [primary outcome] with an estimated standard deviation of [insert standard deviation]. A statistical power of [insert power, e.g., 80%] and a significance level (alpha) of [insert significance level, e.g., 0.05] were used. A [one-/two-]tailed test was chosen based on [reason for choice, e.g., the directionality of the hypothesis]. Based on these assumptions, the required sample size was calculated to be [insert number] participants per group, accounting for a potential dropout rate of [insert dropout rate], leading to a final sample size of [insert final sample size].” For further guidance, you might look at Eng J. Sample size estimation: how many individuals should be studied? Radiology. 2003 May;227(2):309-13 for more details.
Response:
Thank you so much for pointing out this important issue. We totally agree the suggested example for a primary outcome presented by continuous value (mean+/- SD). In this study, the primary outcome was presented by dichotomous value (%). We revised the sample size calculation according the recommendation.
Manuscript Changes:
The sample size was determined based on an anticipated incidence of 47.5% in the primary outcome, which is the moderate and severe catheter-related bladder discomfort (CRBD) [14]. The estimated decrease of CRBD was 30%. A statistical power of 80% and a significance level (alpha) of 0.05 were used. A two-tailed test was chosen based on the bidirectional nature of our hypothesis. Based on these assumptions, the required sample size was calculated to be 37 participants per group, accounting for a potential dropout rate of 5%, leading to a final sample size of 40.
- Table 2 is not very intuitive for interpreting the study results. While it's acceptable to note in the text that there is a significant difference between low- and high-grade CRBD in the two groups, for a primary endpoint, I would expect to see the scores for each grade (Grade 0, 1, 2, and 3) at different time points displayed in one or multiple graphs. This visual presentation would be more effective than a table.
Response:
Thank you so much for pointing out this important issue.
We displayed the scores for each grade (Grade 0, 1, 2, and 3) at different time points using a graph. We could not show statistical significance with 4 grades by ANOVA method. Therefore, we summarize the results into Table 2 to highlight significant differences and enhance clarity.
- The authors explain that “The analgesia nociception index (ANI) is specifically designed to correlate with nociceptive pain” and then clarify that bladder discomfort is classified as “visceral pain” rather than nociceptive pain. This, they suggest, might explain the lack of ANI score correlation in their study. To me, this indicates that ANI is not an appropriate tool for monitoring nociception in cases of bladder discomfort. It is then unclear why ANI was chosen for this study. Could the authors please elaborate on the rationale behind using ANI despite this apparent discrepancy in their Discussion section?
- Response:
We appreciate this important issue.
The reasons for using ANI during design this RCT:
1) ANI provides a non-invasive, real-time assessment of analgesia/nociception balance during anesthesia, which could still offer valuable insights even if not directly correlated with visceral pain.
2) There may be limited alternative options for objectively monitoring nociception in anesthetized patients, and ANI represents one of the more promising tools available.
3) ANI may still indirectly reflect changes in nociception related to bladder discomfort, even if not a direct measure of visceral pain.
Therefore, we chose ANI for its potential to provide objective real-time monitoring of pain perception.
Though our findings did not show significance between ANI score and CRBD, the use of ANI suggested that patients in both groups were under comparable well nociceptive pain control.
According to major comment 3, we added rationale using ANI in the discussion
Manuscript Changes:
The rationale for intra-operative and post-operative ANI monitoring includes the following points: First, ANI provides a non-invasive, real-time assessment of the analgesia/nociception balance during anesthesia, offering valuable insights even if it does not directly correlate with visceral pain. Second, there are limited alternative options for objectively monitoring nociception in anesthetized patients, and ANI is one of the more promising tools available. Third, ANI may still indirectly reflect changes in nociception related to bladder discomfort, even if it is not a direct measure of visceral pain. Despite the apparent discrepancy between ANI findings and CRBD, the use of ANI suggested that patients in both groups were under comparable nociceptive pain control from anesthesia induction to recovery in the PACU.”
- I suggest that the authors provide a summary of the safety assessments conducted in their study and discuss their findings in detail. Specifically, did they monitor for symptoms such as nausea, vomiting, dizziness, tongue tingling, metallic taste, allergic reactions, and so on? If these assessments were performed, what were the incidence rates in the two treatment groups, and were there significant differences? This information is essential to substantiate the authors' conclusion that “Lidocaine bladder irrigation [...] is a practical method with minimal side effects.”
- Response:
We appreciate this important issue. Except pain assessment, we routinely conducted comprehensive safety assessments of postoperative nausea and vomiting (PONV), dizziness, allergic reactions and systemic toxicity of local anesthetics (in this study) such as tongue tingling, metallic taste or seizure. The safety profile was very high in each group and statistical analyses were performed to identify any significant differences. Detailed description was added into least part of outcome measurements and results (after Table 5) respectively. There findings of safety assessments are provided to substantiate our conclusion that lidocaine bladder irrigation is a practical method with minimal side effects.
Manuscript Changes:
Method- Outcome Measurements
We also recorded comprehensive safety assessments including postoperative nausea and vomiting (PONV), dizziness, allergic reactions and any symptom of systemic toxicity of lidocaine (tongue tingling, metallic taste or seizure).
Results
The incidence of PONV was 2 out of 39 in the lidocaine group and 1 out of 40 in the control group (P=0.982). Dizziness was reported in 2 out of 39 in the lidocaine group and 0 out of 40 in the control group (P=0.463). No patients in either group exhibited any allergic reactions or symptoms of lidocaine toxicity.
Minor comments:
- Please include additional information on the primary endpoint, catheter-related bladder discomfort (CRBD), in the Abstract. For example: “The primary endpoint was the assessment of the incidence and severity of CRBD upon awakening within the first 6 hours postoperatively, using a four-grade scale based on patients' reports of discomfort.”
- Response:
We thank you for this comment and revised the description of abstract as ” The primary endpoint was the assessment of the incidence and severity of CRBD upon awakening within the first 6 hours postoperatively, using a four-grade scale based on patients' reports of discomfort.”
. We will consider this protocol in the future trial.
- In the Introduction, please correct the following sentence: “The pathophysiology of CRBD is indeed primarily attributed to the bladder irrigation might irritate the bladder mucosa and activate the muscarinic receptors.” I believe it should read: “The pathophysiology of CRBD is primarily attributed to bladder irrigation, which might irritate the bladder mucosa and activate the muscarinic receptors.”
- Response:
We appreciate your suggestion and revised this sentence of introduction as “The pathophysiology of CRBD is primarily attributed to bladder irrigation, which might irritate the bladder mucosa and activate the muscarinic receptors.”
- In the background subsection of the Abstract, the drug name "lidocaine" is written with a capital letter as “0.05% Lidocaine”. Please correct it to “0.05% lidocaine”.
- Response:
Thank you very much. We correct this capital letter as comment.
- In Section 2.2 Outcome Measurements, "catheter-related bladder discomfort" is written out, even though the abbreviation "CRBD" has been previously used. The same applies to “analgesia nociception index (ANI)” at different places throughout the manuscript.
- Response:
We appreciate this comment and ensure the consistent use of abbreviations of CRBD and ANI.
- In the Discussion section, it is mentioned that “To our knowledge, only limited RCTs assessed the effect of lidocaine irrigation on CRBD [15].” Do the authors mean “[…] only a limited number of RCTs”? If not, please clarify why the referenced RCT (number 15 in the reference list) is of limited quality.
- Response:
Thank you for this comment. We appreciate the referenced RCT to assess effect of lidocaine irrigation on CRBD. We mean quality of those work are very well but only a few investigations focus on this issue. The description was revised as “To our knowledge, only a limited number of randomized controlled trials (RCTs) have assessed the effect of lidocaine irrigation on CRBD [15]”
- The citations for the works of Kim et al. and c et al. are incorrectly formatted in the Discussion section: “(citing reference attached Doo-Hwan Kim, 2020; A. Agarwal, 2008;)”. Please correct this.
- Response:
We appreciate this suggestion. We have revised the citation format.
- There are inconsistencies in the use of spacing before units. For example, incorrect forms such as “20mins”, “25ml”, and “1μg/kg” are used, while correct spacing is seen in “80 years”, “6 h”, and “0.5 mcg/kg”. Please ensure a space is inserted before the unit throughout the manuscript. Additionally, please use either “μg” or “mcg” consistently.
- Response:
We thank you for this suggestion. We correct forms such as “20 mins”, “25 ml”, and “1 μg/kg”. We ensure a space is inserted before the unit throughout the manuscript, and use mcg consistently.
Comments on the Quality of English Language
There are inconsistencies in terminology and formatting that need to be addressed. Additionally, the authors sometimes use articles like 'an' and 'a' incorrectly, either including them unnecessarily or omitting them where they are required. It is recommended that a skilled English writer proofread the manuscript for these errors.
- Response:
We appreciate this important issue. We shall arrange further English editing via the Journals platform.

Round 2
Reviewer 2 Report
Comments and Suggestions for Authors
Thank you for revising the manuscript according to the reviewers’ comments. Please note that on page 5 of the manuscript, the following text from the peer review report is erroneously inserted: “Table 2 is not very intuitive for interpreting the study results. While it's acceptable to note in the text that there is a significant difference between low- and high-grade CRBD in the two groups, for a primary endpoint, I would expect to see the scores for each grade (Grade 0, 1, 2, and 3) at different time points displayed in one or multiple graphs. This visual presentation would be more effective than a table.” Please correct. I have no other comments otherwise.
Author Response
Comment 1:
Thank you for revising the manuscript according to the reviewers’ comments. Please note that on page 5 of the manuscript, the following text from the peer review report is erroneously inserted: “Table 2 is not very intuitive for interpreting the study results. While it's acceptable to note in the text that there is a significant difference between low- and high-grade CRBD in the two groups, for a primary endpoint, I would expect to see the scores for each grade (Grade 0, 1, 2, and 3) at different time points displayed in one or multiple graphs. This visual presentation would be more effective than a table.” Please correct. I have no other comments otherwise.
Response 1:
We deeply thank you so much for providing many important comments to improve our article. The erroneous insertion was removed from the text.
